# Gas/Liquid Chromatography–Mass Spectrometry Analysis of Key Functional Substances Regulating Poll Gland Secretion in Male Camels during Seasonal Estrus

**DOI:** 10.3390/ani13122024

**Published:** 2023-06-18

**Authors:** Lijun Dai, Bao Yuan, Bohao Zhang, Wenli Chen, Xixue Yuan, Xinhong Liu, Yuan Gao, Yong Zhang, Quanwei Zhang, Xingxu Zhao

**Affiliations:** 1College of Life Science and Biotechnology, Gansu Agricultural University, Lanzhou 730070, China; d1312496231@163.com (L.D.); 17693462916@163.com (B.Y.); c293840952@163.com (W.C.); yuanxixue0525@163.com (X.Y.); gaoy@gsau.edu.cn (Y.G.); zhangy@gsau.edu.cn (Y.Z.); 2Gansu Key Laboratory of Animal Generational Physiology and Reproductive Regulation, Lanzhou 730070, China; zhangbhgs@163.com (B.Z.); 13359486884@163.com (X.L.); 3College of Veterinary Medicine, Gansu Agricultural University, Lanzhou 730070, China

**Keywords:** Bactrian camel, poll glands secretion, seasonal estrus, GC/LC-MS, steroids, dopamine

## Abstract

**Simple Summary:**

A gas/liquid chromatography–mass spectrometry (GC/LC-MS) dual platform was used to determine the chemical composition of poll gland secretions from the neck manes of male Bactrian camels during estrus, and then steroids and neurochemicals were identified as the key functional substances for regulating seasonal estrus in male camels. The results make a significant contribution to the literature because despite the low reproductive performance of these economically and scientifically important animals, studies on the chemical composition and physiological functions of poll gland secretions in Bactrian camels are very rare. This is the first study to employ a GC/LC-MS dual platform to provide an unbiased global metabolite profile of male Bactrian poll gland secretions. Our findings provide important insights into the development and function of poll glands in Bactrian camels.

**Abstract:**

Increased poll gland secretion is a major characteristic and indicator of estrus in male Bactrian camels; however, research on these poll glands and their secretion is extremely rare. In this study, we determine the chemical composition of poll gland secretions and identify the key functional substances that regulate seasonal estrus in male camels. A GC/LC-MS dual platform was used to analyze ventral hair (control) and neck mane samples containing poll gland secretions from male Bactrian camels during estrus. Multidimensional and single-dimensional analyses were used to screen differentially expressed metabolites (DEMs) between groups. Functional prediction of enriched metabolites was performed using a Human Metabolome Database comparison and Kyoto Encyclopedia of Genes and Genomes pathway enrichment analysis, which were then compared with a behavioral analysis of male Bactrian camels in estrus. A total of 1172 DEMs and 34 differential metabolic pathways were identified. One metabolite group was found to relate to steroid synthesis and metabolism, and another metabolite group was associated with neural metabolism. Therefore, we speculate that steroids and neurochemicals jointly regulate estrous behavior in male Bactrian camels, thus providing theoretical insights into the development and function of poll glands in Bactrian camels.

## 1. Introduction

The Bactrian camel (*Camelus Bactrianus*) is a unique and dominant species that plays a vital role in desert and semi-desert areas and has important economic and scientific value [1]. However, the average annual growth rate of domestic Bactrian camels over the last 10 years has been less than 5%, with wild Bactrian camels currently listed on the “International Union for Conservation of Nature Red List of Threatened Species” [2]. This is mainly attributed to the unique reproductive mechanism of Bactrian camels, which are induced ovulators that exhibit seasonal estrus [3]. In addition, the natural conditions of their habitat are harsh, and the development of reproduction-related technology is relatively slow, resulting in low reproductive performance [4]. The unique physiological anatomy of the Bactrian camel may correspond to its unique breeding mechanisms. That is, poll glands are present only in male Bactrian and dromedary camels (*Camelus dromedarius*) and appear to degrade following castration [5]. These glands are located on the skin on both sides of the first neck vertebra behind the occiput, where the skin is raised and the mane is long and sparse. The parenchyma of the glands is located in the dermis and comprises multiple almond-colored pyramidal lobules [6,7]. The glands atrophy in the summer non-estrus season, whereas the glandular lobules show cyst-like structures during winter estrus and emit a foul-smelling viscous fluid through the skin covering the glands, which becomes dark brown or black upon exposure to air and attaches to the neck mane [8,9]. This secretion is called poll gland secretion or occipital gland secretion, and herdsmen habitually call it “Bao ke”, which is a transliteration of the Mongolian word “Bokhi” [10].

During the estrus season, male Bactrian camels show sexual desire and other external characteristics (foaming at the mouth, making a “beep” sound, grinding teeth, “beating the water whip”, running, competing for females, itching, loss of appetite, etc.). The enhanced function of poll glands and increased secretion are an evident features of male camels in estrus, and they can be used to measure the degree of estrus [11]. Male camels in estrus often tilt their heads back, rub the pillow neck at the front peak, smear the secreted mucus at the front peak, and emit a specific smell. These behaviors demonstrate the instinct of male camels in estrus to spread their gland secretions. However, studies on the chemical composition and physiological functions of poll gland secretions in Bactrian camels are rare. Some hypothesize that poll gland secretions may be a mixture of sweat and sebaceous glands [12]. For example, samples of male Bactrian camel mane containing poll gland secretions were analyzed, along with pure secretions from the poll glands, using the radioimmunity method and ion chromatography analysis to determine the composition of poll gland secretion [5]. The results showed that the poll gland secretions contain many chemical substances, which can be divided into two main categories: (1) steroid hormones (including progesterone, estrogen, and testosterone) and (2) abundant short-chain fatty acids (including acetic acid, propionic acid, isobutyric acid, n-butyric acid, and isovaleric acid). The latter category may also contain large numbers of inorganic ions; however, this has not been quantified. Ayorinde and Wheeler conducted gas chromatography (GC) analysis of poll gland secretions in Bactrian camels, and reported no significant change in the composition or content of steroid hormones among samples; however, the composition of acid substances changed greatly, which may be attributed to individual differences, sample volatility, sample collection methods, and time [13]. In addition, sex pheromones are “chemical messengers” produced by the body that affect other individuals through their sense of smell. According to chemical composition analysis, the effective extract of sex pheromones is a mixture of acetic acid, propionic acid, isobutyric acid, n-butyric acid, and isovaleric acid [5]. Sex pheromone production is closely associated with the levels of sex hormones. Indeed, the level of androgens in the secretion of male camel glands relates to the level of androgens in the blood serum [14,15], with serum androgen levels increasing as poll gland secretion increases. Moreover, male camel estrous behavior disappears when poll gland secretion disappears, and the male camel glands also degenerate after castration. Given that the chemical composition of poll gland secretion is similar to that of sex pheromones, it is speculated that male camel glands secrete a pheromone whose function is to stimulate and induce estrus in female camels [5].

Few studies have analyzed the function of poll gland secretions from the perspectives of histology, physiology, and biology; therefore, their chemical composition and function must be comprehensively determined and tested. In this study, we employ GC–mass spectrometry (GC-MS) to detect volatile and heat-stable compounds, as well as liquid chromatography–mass spectrometry (LC-MS) as a broad-spectrum technique to detect non-volatile, heat-unstable, and high-/low-molecular-weight compounds. We obtain an unbiased global metabolite profile by measuring poll gland secretions in the neck mane (NM) of male Bactrian camels in estrus. Ventral hair (VH), which is located on the belly of the camel and does not contain any secretions, is used as a control. We then comprehensively identify small molecule metabolites and compare the characteristics of DEMs between VH and NM groups to determine the chemical composition and potential functions of poll gland secretions. We reveal the key functional substances regulating seasonal estrus in male camels. If the key functional substances or sex hormones regulating seasonal estrus can be isolated from the poll gland secretions of male camels, it will promote the Bactrian camels to estrus in advance and stimulate the sexual impulse, and it will provide a theoretical basis for the induction of estrus before the initial estrus, anestrus season, and puerperal state, which has important significance in production practice and scientific research.

## 2. Materials and Methods

### 2.1. Animal Use

All animal samples were collected in strict accordance with the Animal Ethics Regulations (Code: GSU-LC-2020-39) approved by the Animal Protection Committee of Gansu Agricultural University. Male Bactrian camels used in this study were obtained from a farm in Zhangye City, Gansu Province. All camels were free roaming and free feeding during the non-breeding season (April–November) and captive during the breeding season (December–March). Experimental samples were collected from male Bactrian camels in estrus in January. Male camels of similar age (8 years old), weight (about 480 kg), and estrus status (foaming at the mouth, making a “beep” sound, grinding teeth, “beating the water whip”, running, competing for females, itching, loss of appetite, etc.) were selected (n = 6), with none showing reproductive abnormalities. VH and NM stained with poll gland secretions were collected from each male camel for untargeted metabolomic analysis. The design of this study is shown in Figure 1.

### 2.2. Collection and Processing of Samples

Samples were labeled in sterile centrifuge tubes and transported to the laboratory with liquid nitrogen and stored at −80 °C for metabolomic analysis. The collected samples were added to 20 μL of pre-cooled methanol, left for 2 min, ground, and ultrasonically extracted with chloroform in an ice-water bath for 10 min. After standing at low temperature for 30 min, the samples were centrifuged at 13,000 rpm for 10 min at 4 °C. The supernatant was then transferred to a new centrifuge tube for GC-MS metabolomic analysis. The supernatant after centrifugation was filtered through a 0.22 μm organic-phase pinhole filter for LC-MS metabolomics analysis [16]. Quality control (QC) samples were prepared by mixing the extracts of all samples in equal volumes and evaluating the stability of the system MS platform throughout the experiment.

### 2.3. Metabolite Detection

Male camel mane samples were used to identify different metabolite levels. Samples (n = 12) were analyzed by untargeted metabolomics using a 7890B-5977A GC-MS instrument (Agilent J&W Scientific, Folsom, CA, USA) to generate a total ion chromatogram [17]. The gas chromatograph was equipped with a B-5MS capillary column (30 m × 0.25 mm × 0.25 μm), the carrier gas was high-purity helium, the flow rate was 1.0 mL/min, and the temperature of the injection port was 260 °C. The injection volume was 1 μL, no split injection was performed, and the solvent was delayed for 5 min. The temperature program is shown in Appendix A. Sample mass spectrum signal acquisition was based on quadrupole mass spectrometry; the scanning mode was full scan mode (SCAN), and the mass scanning range was *m*/*z* 50–500.

Samples (n = 12) were analyzed using an LC-MS system consisting of an ACQUITY UPLC I-Class Plus ultra-high-performance liquid chromatography tandem QE Plus high-resolution MS to generate a base peak chromatogram (BPC) [18]. Data were obtained using the following chromatographic parameters: column using ACQUITY UPLC HSS T3 (100 mm × 2.1 mm, 1.8 um). The mobile phases A and B were water and acetonitrile, respectively, at a flow rate of 0.35 mL/min. The injection volume was 2 μL, and the elution gradient is listed in Appendix A. Positive and negative ion scanning modes were used to acquire the mass spectrum signal. The mass spectral parameters are listed in Appendix A.

### 2.4. Metabolome Data Collection

Combined with the metabolomics data qualitative software MS-DIAL, qualitative and relative quantitative analyses of the original GC-MS data were performed, and the original data were standardized and preprocessed [19]. MS-DIAL was used for the data-independent acquisition-based qualitative and quantitative analysis of small molecules by mass spectral deconvolution, which efficiently determines precursor ion peaks by analyzing two consecutive data axes. The MS2Dec algorithm was used to extract the “model peaks” in the chromatogram, and the similarity was matched with the public database to qualitatively analyze compounds. Internal standards and QC samples were used for quality control. Internal standard peaks and any known false-positive peaks in the original data matrix were removed, and missing values were replaced with half the minimum value. For each sample, the signal intensity (peak area) of all peaks was segmented and normalized according to the internal standard, with a relative standard deviation of <0.3 after screening.

Combined with the metabolomics data processing software Progenesis QI v2.3 (Nonlinear Dynamics, Newcastle, UK), we performed qualitative and relative quantitative analyses of the original LC-MS data, and then we standardized and preprocessed the original data [20]. Progenesis QI v2.3 software was used to extract and process the feature peaks of the original data, QC samples were used for quality control of the data, and ion peaks with a relative standard deviation of >0.3 in the QC group were deleted. Compounds were identified based on their accurate mass number, secondary fragmentation, and isotopic distribution, and qualitative analysis was performed using the Human Metabolome Database (HMDB, https://hmdb.ca/, accessed on 1 October 2022), Lipidmaps (v2.3), and a metabolite mass spectral (METLIN, http://metlin.scripps.edu, accessed on 1 October 2022) database. Finally, the data for positive and negative ions were combined, and subsequent analyses were performed.

### 2.5. Metabolome Data Analysis

Supervised orthogonal partial least squares discriminant analysis (OPLS-DA) was performed to distinguish the overall differences in metabolic profiles among the groups. To prevent overfitting of the OPLS-DA model, seven-fold cross-validation and 200 response permutation testing (RPT) were used to examine the quality of the model [16,17,18]. After quality control, the metabolites detected in male Bactrian camel samples (six VH and six NM samples) were analyzed using a combination of multidimensional and single-dimensional analyses to screen for differential metabolites between groups [18,19,20]. Variable importance in projection (VIP) values obtained from the OPLS-DA model can be used to measure the influence of strength and the explanatory power of metabolite expression patterns on the classification and discrimination of VH and NM samples. Student’s T-test and fold-change analysis were used to verify whether differential metabolites between VH and NM groups were significant. The screening criteria were *VIP* > 1 for the first principal component of the OPLS-DA model and *p* < 0.05.

### 2.6. Metabolic Pathway Analysis

Pathway enrichment analysis of the DEMs was conducted to identify the mechanisms underlying metabolic pathway changes in poll gland secretions. Enrichment results for metabolic pathways were obtained using the Kyoto Encyclopedia of Genes and Genomes (KEGG) ID pathway enrichment analysis of DEMs [21]. Hypergeometric tests were used to identify pathway entries that were significantly enriched in significant DEMs compared to the entire background. Taking *p* ≤ 0.05 as the threshold, the pathway that satisfied this condition was considered to be significantly enriched in differential metabolites; the smaller the *p*-value, the more significant the difference in the metabolic pathway.

## 3. Results

### 3.1. Overview of GC-MS Detection

To identify volatile and thermally stable metabolites in the VH and NM groups, GC-MS was coupled with the detection of unknown components in the secretions of male Bactrian camel glands for qualitative and quantitative analysis. Within the selected mass range, the sum of all ionic strengths exhibited significant differences in terms of time or scanning times, indicating a large difference in the type and content of metabolites between the two groups. A total of 467 metabolites were obtained by MS (Figure 2A,B). The stability of the MS system was analyzed and evaluated through QC of the internal standard and QC samples. The spectral overlap comparison of the total ion chromatogram of QC samples showed an overlap of the response intensities and retention times of the chromatographic peaks, indicating minimal variation caused by instrumental errors during the experiment (Figure 2C). The metabolite intensity distributions of VH, NM, and QC samples were evaluated to observe the overall distribution among the samples, as well as the stability of the entire analysis process. No abnormal samples were found, and the stability was good (Figure 2D). In the OPLS-DA score plot, the predicted principal component PC1 directly distinguished the intergroup variation of VH and NM groups, whereas the orthogonal principal component Pco1 reflected the intragroup variation of VH and NM groups. We identified significant differences in the OPLS-DA score plot between the two groups (Figure 2E). The quality of the model was investigated using seven-fold cross-validation and 200-fold RPT. The interpretive rate R^2^Y(cum) and predictive rate Q^2^(cum) were 1 and 0.995, respectively. As both were greater than 0.9, the model had high predictive ability and a good fitting degree. The parameters R^2^ and Q^2^ of the RPT were 0.894 and −0.523, respectively, indicating that the OPLS-DA model was well established and effectively available (Figure 2F).

### 3.2. Overview of LC-MS Detection

To identify less volatile or less thermally stable metabolites in the VH and NM groups, LC-MS was coupled with the detection of unknown components in the secretions of male Bactrian camel glands for qualitative and quantitative analysis. Primary and secondary mass spectroscopic information on the metabolites was collected, and the intensity of the strongest ions in the mass spectrum at each time point was recorded to obtain a BPC. A total of 29,087 substance peaks and 13,658 metabolites were detected, among which 12,244 substance peaks and 6428 metabolites were identified by positive ion patterns (Figure 3A,B). The negative ion pattern identified 16,843 material peaks and 7230 metabolites (Figure 3C,D). Similarly to GC-MS analysis, the stability of the MS system was analyzed and evaluated by quality control of the QC samples. According to the principal component analysis model diagram obtained by seven-fold cross-validation, the QC samples were closely clustered together, indicating that the instrument had good detection stability during the experiment (Figure 3E). A boxplot was constructed for the metabolite intensity of samples in the VH, NM, and QC groups, where the Y-coordinate is the log10 value of the mass spectral intensity. The boxplot indicated a large fluctuation between VH and NM groups, with notable differences between the two (Figure 3F). Supervised OPLS-DA was performed to distinguish the overall differences in the metabolic profiles between groups, and all samples fell within the 95% confidence interval (Figure 3G). The interpretive rates R^2^Y(cum) and predictive rates Q^2^(cum) obtained by the RPT were 0.999 and 0.995, respectively, both of which were greater than 0.9, indicating that the model had a high predictive ability and a good fitting degree. The parameters R^2^ and Q^2^ of the RPT were 0.677 and −0.62, respectively, indicating that the accuracy derived by random ranking of the OPLS-DA model and by avoiding the classification obtained by the supervised learning method was not accidental (Figure 3H).

### 3.3. DEMs Identified by GC/LC-MS

After quality control, the metabolites detected in male Bactrian camel samples (six VH and six NM samples) were analyzed using a combination of multidimensional and single-dimensional analyses to screen the DEMs between groups. A total of 171 and 1008 DEMs were screened from the 467 and 13,658 metabolites detected by GC-MS and LC-MS, respectively. Hierarchical clustering was conducted to determine the expression of the DEMs (Figure 4A,B). The *p*-value, fold-change value, and VIP value were visualized using a volcanic map, whereby 152 upregulated and 19 downregulated DEMs were detected using GC-MS, and 653 upregulated and 355 downregulated DEMs were detected by LC-MS (Figure 4C,D).

To determine the biological significance of the DEMs, the DEMs selected from the GC-MS and LC-MS dual platforms were pooled, and seven metabolites (sucrose, gentisic acid, cyanuric acid, rhein, isobutyrylglycine, indoxyl sulfate, and propionylglycine) were detected using both platforms. After deleting the duplicated metabolites, 1172 DEMs were obtained between the VH and NM samples of male Bactrian camels in estrus and classified according to their chemical structures (Figure 4E). Considering the qualitative information of metabolites or compound types, DEMs with KEGG orthology numbers were the main DEMs. In the 18 DEM classification groups, many DEMs were enriched in benzenoids, lipids, lipid-like molecules, organic acids and derivatives, organic oxygen compounds, and organoheterocyclic compounds.

### 3.4. Analysis of DEMs

A total of 1172 DEMs were identified by GC/LC-MS. Compound information was compared in the HMDB according to 18 structural categories to determine the function and interaction relationships of each metabolite. After comparing all possible metabolite functions, significant enrichment was identified in two groups of metabolites, i.e., 48 neural metabolites (Appendix A) and 19 steroid metabolites (Appendix A), suggesting that poll gland secretion in male Bactrian camels during estrus may be regulated by neurohormones (Figure 5A,B). More rigorous difference values were used to select the biologically significant DEMs. Considering that the absolute log2(FC) value of the neural metabolites was large and that this group formed the majority, we further determined that most of the 48 neural metabolites were neurotransmitters. According to their chemical properties, we classified these neural metabolites into acetylcholine, amino acids, catecholamines, indoleamine, Purine, and Other (Figure 5C). Volcanic plots of the neural metabolites contained in these six groups revealed four downregulated and 44 upregulated metabolites, with high log2(FC) values of 37.38 and 38.02 for adenylosuccinate and sodium gluconate, respectively (Figure 5D). In addition, catecholamines accounted for the largest proportion, and their functions relate to dopamine synthesis and metabolism, which may be the main regulatory hub with biological significance. Information regarding these 17 metabolites is shown in Figure 4E, which indicates that all compounds have a benzene ring structure, except 3-acetamidopropanal.

### 3.5. Analysis of Differential Metabolic Pathways

KEGG pathway enrichment analysis and hypergeometric tests were used to screen 34 pathway items, with *p* < 0.05. To visualize the number of DEMs in the pathway and the significance between the enrichment factors, a bubble chart was drawn to highlight the significantly enriched pathways, of which nine related to neural metabolism (Figure 6A). Based on the Cytoscape network topology parameters, a KEGG network diagram of nine neuro-related metabolic pathways was constructed to show the relationship between enriched pathways and metabolites (Figure 6B), where the size of the node represents the number of metabolites enriched in the pathway after KEGG enrichment analysis, and the gradient color of the node represents the *p*-value of the KEGG enrichment analysis. In this network, 26 DEMs were enriched after the addition of the nine connecting metabolic pathways. According to the direction and degree of association between different metabolic nodes, three metabolic pathways (tyrosine metabolism, tryptophan metabolism, and dopaminergic synapses) and three DEMs (DL-dopa, 3,4-Dihydroxybenzeneacid, and L-glutamic acid) were identified as the core metabolic nodes. The results suggest that dopamine, the main neurotransmitter in the extrapyramidal system of the brain, plays an important role in regulating the enhancement of poll gland function and seasonal estrus in male Bactrian camels. The 26 DEMs enriched in the neural metabolism pathway after KEGG enrichment analysis were drawn as volcano maps and heat maps to show the differentially connected metabolites between VH and NM groups, among which 4 were downregulated, and 22 were upregulated (Figure 6C,D).

### 3.6. Determine the Functional Prediction of Metabolites

To further explore the correlation between poll gland secretion and neural metabolites in the differential samples of VH and NM groups, 17 catecholamines were sorted and screened from the 1172 DEMs, 26 neural node metabolites enriched from 9 neuro-related metabolic pathways were used for the Venn diagram analysis, and a total of 10 co-expressed DEMs were obtained (Figure 7A,B). A dotted heatmap was drawn to show both the metabolite VIP values and expression abundance trends in different samples, where the maximum VIP value of 3,4-Dihydroxybenzeneacetic acid was 28.09 (Figure 7C). The KEGG metabolic pathway was used to determine the production process of each metabolite, which revealed that the 10 DEMs from the Venn diagram were involved in tyrosine metabolism and related to dopamine synthesis and metabolism. Considering the characteristics of the poll glands and neck mane of male Bactrian camels during estrus, we speculated that dopamine directly or indirectly regulates blackening of the neck mane and enhancement of poll gland functions in male Bactrian camels during estrus.

## 4. Discussion

In this study, we sampled VH and NM containing poll gland secretion from male Bactrian camels in estrus for untargeted metabolomics evaluation based on the differences in metabolites and metabolic pathway analysis, screening function enrichment, and significantly highly expressed metabolites to determine the chemical composition and underlying mechanisms of poll gland secretion. In these samples, 1172 metabolites and 34 metabolic pathways were significantly different, indicating significant metabolic differences between VH and NM groups and showing that both groups contained a large number of metabolites. After comparing all possible metabolite functions using the HMDB database, two groups of metabolites were found to be significantly enriched. One group was represented by testosterone, which related to steroid synthesis and metabolism. The other group, represented by dopamine metabolites, is associated with neural metabolism.

### 4.1. Metabolites Relating to Steroid Synthesis and Metabolism in Poll Gland Secretions

Testosterone, the most important male sex hormone for the development of male reproductive tissues such as the testes and prostate, was highly expressed in the NM group. Like other steroid hormones, testosterone is derived from cholesterol and forms the center of steroid metabolite network connections [22]. Estriol-16-glucuronide is a derivative of estriol, whereas estriol-3-glucuronide is a natural estriol metabolite [23]. Therefore, the upregulation of large amounts of cholesterol and the differential expression of steroid-like metabolites suggest the synthesis of sex hormones in the poll glands of male Bactrian camels. Similarly, previous studies have proposed that the poll glands of male camels are steroid-dependent organs that accumulate anabolic steroids [15,24,25,26]. Research has reported that after the start of the estrus season in male camels, the anterior pituitary secreting cells and Leydig cells become active, and the testicular weight increases. The diameter of the seminiferous tubule can reach 209–220 μm, and the number of sperm per gram of testicular tissue can reach 3.6 to 4.7 million; in the absence of estrus, the diameter of the tubule is 190–203 μm, and the number of sperm per gram of testicular tissue is 2.7–3 million [27]. Furthermore, the concentrations of testosterone, cholesterol, and cortisol in the blood of male camels increased significantly during the estrus season, and the concentration of follicle-stimulating hormone also reached a peak [28], while the level of androgens in the secretion of male camel glands relates to the level of androgens in the blood serum [14,15], with serum androgen levels increasing as poll gland secretion increases. Moreover, male camel estrous behavior disappears when poll gland secretion disappears, and the male camel glands also degenerate after castration, which indicates that poll gland secretions closely relate to sexual activities of male camels.

In addition, methylheptenone acts as an alarm or an attractant in insects, particularly in animals harboring the odorant receptor gene *Or4* [29]. In this study, methylheptenone was overexpressed in the NM group, whereas acetic acid, propionic acid, isobutyric acid, n-butyric acid, and isovaleric acid, which are thought to be effective sex pheromones, were not detected. These compounds were replaced with 2-hydroxybutanoic acid, alpha-aminoadipic acid, butanedioic acid, (S)-2-hydroxyglutarate, oxoglutaric acid, aminomalonate, hydroxypropanedioic acid, and 2,6-diaminopimelic acid. Sex pheromones provide information about sex, reproductive status, individual identity, competitiveness, and health status [30]. Thus, isolating the sex pheromones or synthetic pheromones of functional groups from the poll gland secretions of Bactrian camels could be used to stimulate their sexual impulses in advance, which has considerable significance for scientific research and breeding efforts. Most sex hormones exist in a mixed state, and the structure and function of the production sources or organs of sex hormones show seasonal changes, species differences, and a correlation with ontogeny [31]. However, currently available information is based on a limited number of Bactrian camels, and it remains unclear whether poll gland secretions contain typical sex pheromones. Therefore, future research should systematically collect and examine large numbers of animal samples throughout the year.

### 4.2. Metabolites Relating to Neural Metabolism in Poll Gland Secretions

The total thickness of the skin and dermis at the poll glands of male Bactrian camels is greater than that of other skin on the neck. The dermis contains hair follicles, blood vessels, lymphatic vessels, nerves, sebaceous glands, and sweat glands, whereas the poll glands are located in the dermis, adrenergic axons, and blood vessels, including sinusoidal capillaries present in the intralobular connective tissue close to secretory cells, with the axons containing numerous dense vesicles and sparse mitochondria [7,15,26,32]. Poll gland secretion increases when male camels are stressed or stimulated [15]. Thus, we speculate that the adrenal glands play an important role in the estrous behavior of male Bactrian camels, including the increase in poll gland function. Dehydroepiandrosterone, a natural steroid hormone produced by the adrenal glands, was highly expressed in the NM group, which further supports this finding. Male Bactrian camels enter mating season in a state of sexual excitement, showing extreme sensitivity to the female camel, which herders often term “tide” or “crazy”. Male camels in heat often foam at the mouth (even the whole head may be covered with white foam), urinate frequently, and their urine rhythmically flicks up and down with the tail. The more excited the male camel, the more evident these phenomena are. In addition, after the onset of estrus, male Bactrian camels lose appetite and weight through up to 16–25% abdominal contraction, while retaining their previous energy levels. Animals become more aggressive toward other male camels, exhibiting violence and even biting [11]. Male camel estrus is generally believed to be controlled by androgens [33]; however, according to the highly expressed and significantly enriched DEMs and KEGG pathways identified in the NM group, we suggest a close relationship between the neural and endocrine systems, characterized by bidirectional information transmission and interaction, which regulates the behavior of male Bactrian camels in estrus.

The other group of retrieved metabolites associated with neuro-related metabolic pathways was analyzed using the HMDB database, and the results showed enrichment of a large number of catecholamine metabolites, suggesting that dopamine, a major neurotransmitter in the brain extrapyramidal system, plays an important role in estrus behavior. Dopamine is a member of the catecholamine neurotransmitter family and a precursor of epinephrine and norepinephrine. In vivo, tyrosine is first hydrated to dopa (L-Dopa) by tyrosine hydroxylase, which is then synthesized by aromatic-L-amino-acid decarboxylase [34]. This process occurs mainly in the cytosol of neural tissues and adrenal glands, where the synthesized dopamine is taken up into storage vesicles, released through the synaptic cleft, and actively absorbed from the extracellular space by specific transporters [35]. Dopamine is mainly decomposed by monoamine oxidase (MAO) and catechol-O-methyltransferase (COMT) to generate 3,4-dihydroxybenzeneacetic acid, homovanillic acid, 3,4-dihydroxymandelic acid, and vanillylmandelic acid, all of which were highly expressed in the NM group. Studies have shown that dopamine affects appetite by stimulating exocrine secretion and reducing gastrointestinal motility in the digestive system [36]. In the kidneys, dopamine increases sodium excretion and urine output, leading to frequent urination, whereas high dopamine levels can lead to hyperactivity, anxiety, hypersalivation, and digestive problems [37,38]. Normal concentrations of dopamine in blood vessels can inhibit the release of norepinephrine and act as a vasodilator [39]. Moreover, dopamine is a key neurotransmitter that controls sexual function and is associated with sexual drive and mating performance. Sexual arousal occurs with an increase in dopamine agonists (e.g., L-dopa), which induce genital arousal (i.e., penile erection), promote mating, and lower the ejaculation threshold in male rats [40,41]. In addition, the catecholaminergic system is involved in the regulation of aggressive behavior. Drugs that increase central dopaminergic transmission (e.g., amphetamines and cocaine) are known to trigger psychosis, in which aggressive behavior is violent and uncontrollable. Furthermore, dopamine may reduce the aggressive response threshold to environmental stimuli, with male mice showing knockout of *MAOA* and *COMT* genes exhibiting greater aggression [42]. All of these factors correspond to the estrous behavior of male Bactrian camels.

Thus, by combining knowledge of the estrous behavior and physiological function of dopamine in male Bactrian camels, we speculate that dopamine plays an important role in regulating neck and poll gland cell proliferation and differentiation in male Bactrian camels during estrus (Figure 7D). L-dopaquinone, a metabolite of L-dopa and melanin precursor, was highly expressed in the NM group. L-dopaquinone is highly reactive with other compounds and typically combines with cysteine to form pheomelanin or is converted to L-dopachrome and finally to eumelanin [43]. Homogentisic acid was also detected as a metabolite of tyrosine that relates to the synthesis of pheomelanin, which can be converted to benzoquinone acetic acid to produce polymers similar to skin melanin; however, the difference was not as significant as that of L-dopaquinone. Dopamine receptors are members of the G protein-coupled family, in which dopamine receptors 1 and 5 exhibit stimulatory properties and can be positively coupled to adenylate cyclase [44]. When cells are stimulated by dopamine, dopamine first binds to the dopamine receptor D5 to form a complex, and then it activates guanine nucleotide-binding protein and adenylate cyclase 5 on the cell membrane, catalyzing the removal of pyrophosphate from ATP to generate cAMP, which acts as a second messenger to phosphorylate MAPK pathway proteins by activating PKA, thereby regulating the proliferation, differentiation, and apoptosis of cells [45]. Excess metabolic cAMP is eventually hydrolyzed by phosphodiesterase to AMP, which is then inactivated. The log2(FC) of adenylosuccinate (N6-AMP) in the NM group was 37.38, higher than that in the VH group, which further confirms this hypothesis. In addition, the dopamine receptor D1 can bind to dopamine and activate the proto-oncogene (*c-fos*) through PKC to affect downstream gene regulation [46]. However, further research is required to determine whether increased swelling and activity of male Bactrian camel poll glands in the estrus season, along with atrophy and degeneration in the non-estrus season, relate to the direct or indirect regulation of poll gland cell proliferation, differentiation, and apoptosis by dopamine.

## 5. Conclusions

The aim of this study was to determine the chemical composition of poll gland secretions in male Bactrian camels and reveal the key functional substances regulating seasonal estrus in male camels. The originality of this study lies in our use of a GC/LC-MS dual platform to detect poll gland secretions in male Bactrian camels during estrus, which provided an unbiased global metabolite profile. A total of 1172 DEMs and 34 differential metabolic pathways were screened, and their functions were predicted by comparing the results to estrous behavior in male Bactrian camels. One metabolite group was found to relate to steroid synthesis and metabolism, which suggests that anabolic steroids accumulate in the poll glands; however, the presence of typical sex pheromones was not confirmed in poll gland secretions. Another metabolite group was associated with neural metabolism. Additionally, the high expression of many catecholamine metabolites suggests that dopamine plays a major role in the estrous behavior of male Bactrian camels by directly or indirectly regulating blackening of the neck mane and the proliferation, differentiation, and apoptosis of poll gland cells, including enhanced poll gland function. Some of the discovered metabolites were supported by the previous literature; however, others require further validation to highlight the function of poll gland secretion. By clarifying the composition and function of poll gland secretions, this study also has application for Bactrian camel breeding technology.

## Figures and Tables

**Figure 1 animals-13-02024-f001:**
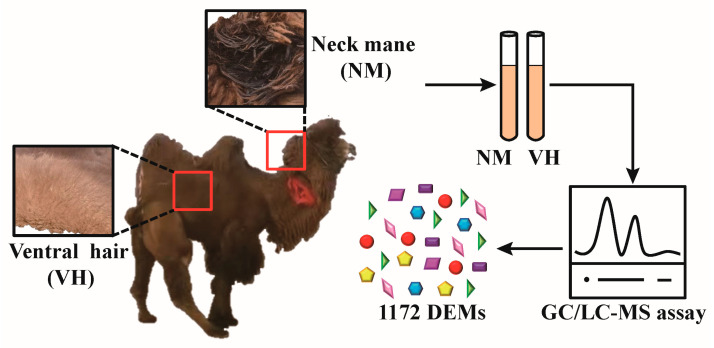
Schematic diagram of the study design.

**Figure 2 animals-13-02024-f002:**
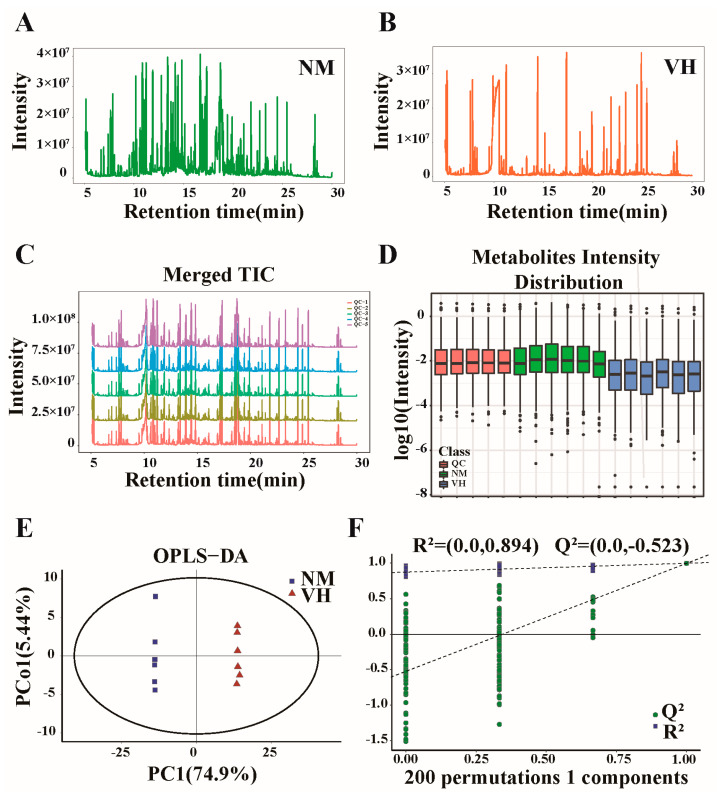
Study design and analysis of GC-MS data. (**A**,**B**) Total ion chromatogram. (**C**) Total ion chromatogram overlay diagram of QC samples. (**D**) Boxplot of metabolite strength. (**E**) Orthogonal partial least squares discriminant analysis (OPLS-DA) score plot. (**F**) Permutation diagram showing the permutation test of OPLS-DA, where R^2^ and Q^2^ values are the parameters for response permutation testing. DEMs, differentially expressed metabolites. NM, neck mane, i.e., the mane at the poll glands in Bactrian camels. VH, ventral hair, i.e., the hair on the belly of Bactrian camels. QC, quality control.

**Figure 3 animals-13-02024-f003:**
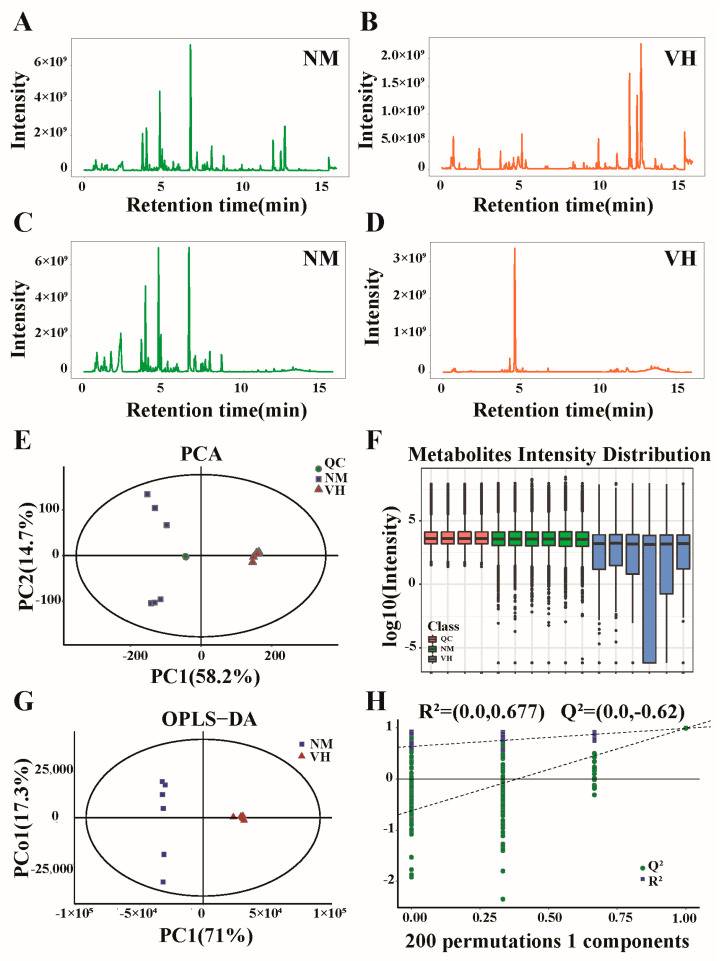
Analysis of LC-MS data. (**A**,**B**) Base peak chromatogram for positive ion pattern and (**C**,**D**) negative ion pattern. (**E**) Principal component analysis score plots for all samples. (**F**) Boxplot of metabolite strength. (**G**) OPLS-DA score plot. (**H**) Permutation diagram showing the permutation test of OPLS-DA. R^2^ and Q^2^ are the parameters for response permutation testing. NM, neck mane. VH, ventral hair. QC, quality control.

**Figure 4 animals-13-02024-f004:**
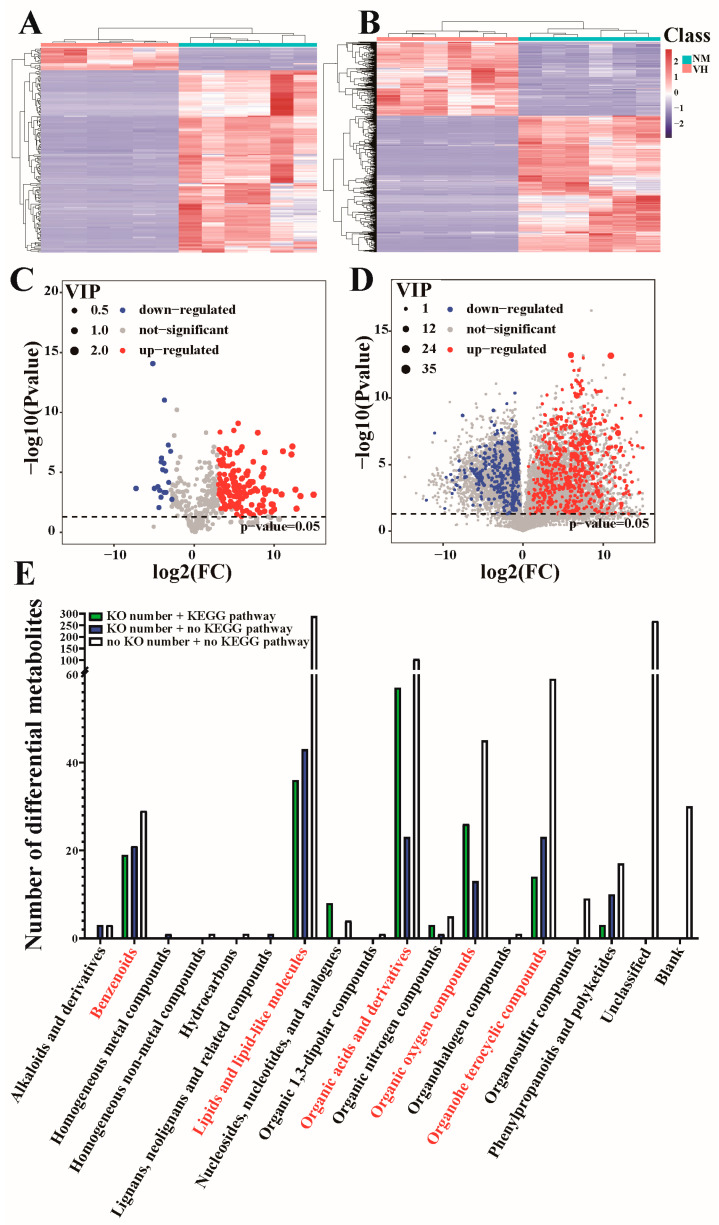
A total of 1172 DEMs were detected by GC/LC-MS and classified. (**A**,**C**) Heatmap and volcano map of 171 GC-MS DEMs. (**B**,**D**) Heatmap and volcano plot of 1008 LC-MS DEMs. (**E**) Chemical structure classification of 1172 GC/LC-MS DEMs. NM, neck mane. VH, ventral hair. VIP, variable importance in projection.

**Figure 5 animals-13-02024-f005:**
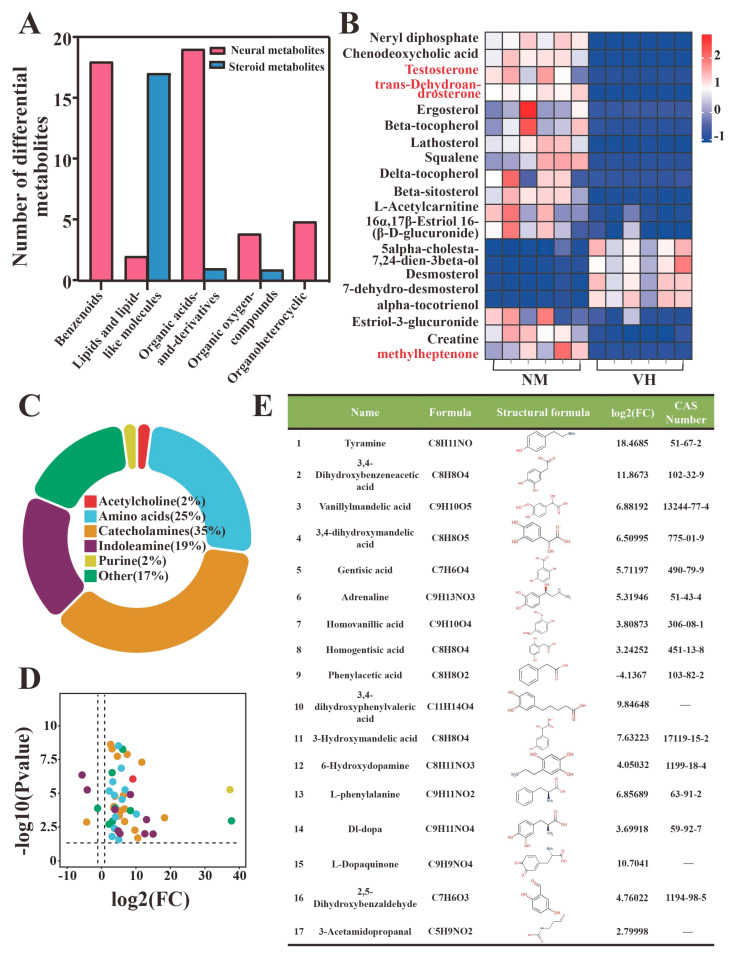
Selection of candidate metabolites relating to neurometabolites from 1172 DEMs. (**A**) According to 18 chemical structural classifications of 1172 DEMs, 48 neural metabolites and 19 steroid metabolites were detected and enriched. (**B**) Heatmap of 19 sterol metabolites. (**C**,**D**) 48 neural metabolites were subdivided into 6 neurotransmitter-related metabolites, and the *p*-value and fold-change value were visualized using the volcano plot. (**E**) Information on 17 catecholamines metabolites. NM, neck mane. VH, ventral hair.

**Figure 6 animals-13-02024-f006:**
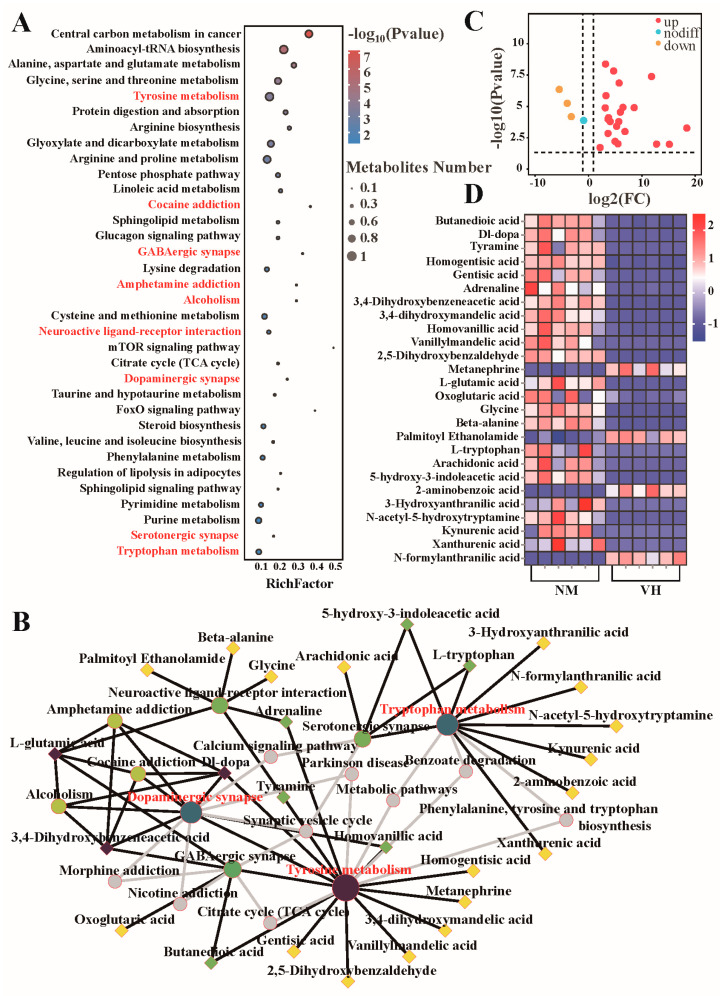
Selection of candidate metabolites from 34 significantly different metabolic pathways relating to nerves. (**A**) Based on KEGG annotations of 1172 DEMs, 9 neuro-related metabolic pathways were selected from 34 significantly different pathways. (**B**) KEGG network map based on the nine neural-related metabolic pathways. ◇ represents metabolites. ○ represents the KEGG pathway. The larger the node graph, the more metabolite enrichment; the darker the node color, the more significant the difference. Gray node indicates the pathway supplemented from the KEGG connection database. (**C**,**D**) Volcano plots and heatmaps of 26 DEMs enriched in neural-related KEGG pathways. NM, neck mane. VH, ventral hair.

**Figure 7 animals-13-02024-f007:**
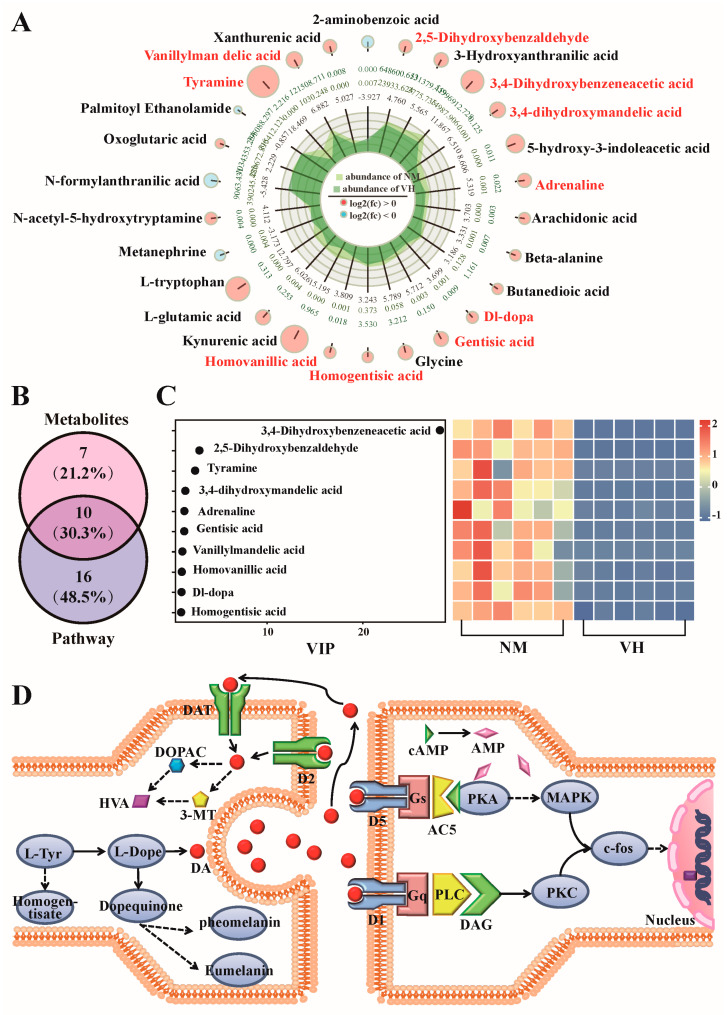
Selection of co-expressed neurometabolites from classified DEMs and KEGG pathways for analysis and functional prediction. (**A**) Report radar diagram of 26 DEMs enriched in neural-related KEGG pathways. (**B**) Venn diagram of co-expressed metabolites associated with neural selection from classified DEMs and KEGG pathways. (**C**) Dotrod heatmap of 10 co-expressed metabolites was obtained from classified DEMs and KEGG pathways. (**D**) Mechanism of action of dopamine in the poll glands of Bactrian camels. NM, neck mane. VH, ventral hair. VIP, variable importance in projection. DA, Dopamine. DOPAC, 3,4-Dihydroxyphenylacetic acid. 3-MT, 3-Methoxytyramine. HVA, Homovanillic acid. D1, dopamine receptor 1. D2, dopamine receptor 2. D5, dopamine receptor 5. DAT, solute carrier family 6 member 3. Gs, guanine nucleotide-binding protein G(olf) subunit alpha. Gq, guanine nucleotide-binding protein G(q) subunit alpha. AC5, adenylate cyclase 5. PLC, phosphatidylinositol phospholipase C. DAG, Diglyceride. PKA, protein kinase A. PKC, protein kinase C. MAPK, mitogen-activated protein kinase. *c-fos*, proto-oncogene *c-Fos*.

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
