# Peer review of "Gas/Liquid Chromatography–Mass Spectrometry Analysis of Key Functional Substances Regulating Poll Gland Secretion in Male Camels during Seasonal Estrus"

_animals, 2023, doi:10.3390/ani13122024_

Round 1
Reviewer 1 Report
The present investigation studies "substances regulating poll gland secretion in male camels during seasonal estrus". This research provides interesting information. However, some important changes need to be made before final publication.
Summary: review the "Journal" guidelines. It is mentioned in "MDPI Style Guide" the following: "The abstract contains a summary of the entire article and can be up to 200 words with a single paragraph." (https://www.mdpi.com/authors/layout) In this case it exceeds the number of words. Therefore, restructure this section.
INTRODUCTION
General comments:
I recommend being more specific with the objective of this study.
Specific comments:
Line 115, 116.- Mention "The design of this study is shown in Figure 1(A)."
MATERIAL AND METHODS
General comments: I recommend mentioning what were the selection criteria for these animals. Also, mention which variables were evaluated.
Line 138.- A message appears "Error! Reference source not found..." check this.
Line 183.- it is mentioned "qualitative analysis was performed using the Human Metabolome Database (HMDB, https://hmdb.ca/)" how do they justify using a human base to compare with camel data?
Line 187.- the section "Statistical analysis" goes at the end of the section "Material and methods".
Line 192.- A message appears "Error! Reference source not found..." check this.
Line 195.- A message "Error! Reference source not found.." check this.
RESULTS
Line 238.- "Figure 1. Study design and analysis of GC-MS data." This goes in the "Material and methods" section. I recommend where it mentions methodology to change it.
Line 352.- in Figure 5A it is mentioned "Based on KEGG annotations of 1,172 DEMs, nine neuro-related metabolic pathways were selected from 34 significantly different pathways" and some risk factors are mentioned among which are; "Cocaine addiction, Anphetamine addiction, Alcoholism" because this is considered. Here it is related to the above mentioned why use a human base?
DISCUSSION
In general, I recommend orienting the discussion according to how you reported your results.
In addition, I recommend being more punctual with the account of your results with the discussion section. As well as expanding this section.
Line 399.- mentions "Testosterone, the most important male sex hormone for the development of male reproductive tissues such as the testes and prostate, was highly expressed in the NM group". If you did not measure testosterone or some other hormone this seems more like speculation. I wish you had related hormones, metabolites and their pathways. Also, I consider that it is not completely correct to relate two different species. Therefore, the overall discussion is not well supported by the results of this research.
CONCLUSION
I recommend restructuring this section and being more specific about the findings mentioned in the results.
Author Response
Comments from the reviewer 1
Summary
Comment 1: Review the "Journal" guidelines. It is mentioned in "MDPI Style Guide" the following: "The abstract contains a summary of the entire article and can be up to 200 words with a single paragraph." (https://www.mdpi.com/authors/layout) In this case it exceeds the number of words. Therefore, restructure this section.
Response: Thank you for your suggestion, we have reduced the number of words in the abstract.
Introduction
Comment 2: I recommend being more specific with the objective of this study.
Response: Thank you for your suggestion. We have revised it based on your suggestion.
Comment 3: Line 115, 116. Mention "The design of this study is shown in Figure 1(A)."
Response: Considering the typography and overall aesthetics of the article, the schematic diagram of the research design is combined in Figure 1 and marked as Figure 1 (A).
Material and methods
Comment 4: I recommend mentioning what were the selection criteria for these animals. Also, mention which variables were evaluated.
Response: Thanks for you bring this suggestion. We have added some details based on your comments.
Comment 5: Line 138. A message appears "Error! Reference source not found..." check this.
Response: Thank you for your suggestion. We have revised it based on your suggestion.
Comment 6: Line 183. it is mentioned "qualitative analysis was performed using the Human Metabolome Database (HMDB, https://hmdb.ca/)" how do they justify using a human base to compare with camel data?
Response: Thank you for your suggestion. The current qualitative database of metabolomics is generally non-species. In the early years, the human metabolome database only included metabolites related to humans, in addition to plants, cattle and so on, and then all merged. Although the latest HMDB 5.0 is still called human metabolic database, it is a non-species database, and the identification process is actually a matching scoring process.
Comment 7: Line 187. the section "Statistical analysis" goes at the end of the section "Material and methods".
Response: Thank you for your suggestion. In metabolomics analysis, the qualitative and quantitative results were analyzed to obtain the differential metabolites between groups, and then the enrichment analysis of metabolic pathways for the differential metabolites was conducted based on KEGG database. After consulting the data, it would be more appropriate to change the title "statistical analysis" to "metabolome data analysis".
Comment 8: Line 192. A message appears "Error! Reference source not found..." check this.
Response: Thank you for your suggestion. We have revised it based on your suggestion.
Comment 9: Line 195. A message "Error! Reference source not found.." check this.
Response: Thank you for your suggestion. We have revised it based on your suggestion.
Results
Comment 10: Line 238. "Figure 1. Study design and analysis of GC-MS data." This goes in the "Material and methods" section. I recommend where it mentions methodology to change it.
Response: Thank you for your suggestion. We have revised it based on your suggestion.
Comment 11: Line 352. in Figure 5A it is mentioned "Based on KEGG annotations of 1,172 DEMs, nine neuro-related metabolic pathways were selected from 34 significantly different pathways" and some risk factors are mentioned among which are; "Cocaine addiction, Anphetamine addiction, Alcoholism" because this is considered. Here it is related to the above mentioned why use a human base?
Response: Thank you for your suggestion. Although the name of HMDB is still human metabolome database, it integrates early plant, animal and other databases, and is the largest and most comprehensive biology-specific metabolite database. Pathway enrichment analysis was conducted using KEGG ID of differential metabolites, and the enrichment results of metabolic pathway were obtained. The smaller the P-value, the more significant the difference of the metabolic pathway. The metabolic pathways of cocaine addiction, amphetamine addiction and alcoholism were significant because significant differentially expressed metabolites detected in the samples were significantly enriched in these pathway entries compared with the whole background.
Discussion
Comment 12: In general, I recommend orienting the discussion according to how you reported your results.
Response: Thank you for your suggestion. As you suggested, we restructured the discussion section as reported results.
Comment 13: In addition, I recommend being more punctual with the account of your results with the discussion section. As well as expanding this section.
Response: Thank you for your suggestion. We have revised it based on your suggestion.
Comment 14: Line 399. mentions "Testosterone, the most important male sex hormone for the development of male reproductive tissues such as the testes and prostate, was highly expressed in the NM group". If you did not measure testosterone or some other hormone this seems more like speculation. I wish you had related hormones, metabolites and their pathways. Also, I consider that it is not completely correct to relate two different species. Therefore, the overall discussion is not well supported by the results of this research.
Response: Thank you for your suggestion. Both testosterone and trans-Dehydroandrosterone were detected by metabonomics, and their content in the NM group was significantly higher than that in the VH group. The content of each sample and the detection data were shown specifically in the supplementary material Table S5, which has been shown in Figure 4 (B) in the paper, so it is not a guess.
Conclusion
Comment 15: I recommend restructuring this section and being more specific about the findings mentioned in the results.
Response: Thank you for your suggestion. We have revised it based on your suggestion.

Author Response
Comments from the reviewer 2
Comment 1: In Simple Summary, a brief description of key findings is suggested, apart from the emphasis on the importance of the study.
Response: Thank you for your suggestion. We have changed it based on your suggestion.
Comment 2: Are there any advantages in collection of poll gland secretions using neck mane over directly absorbing secretions using gauze? The latter one probably leads to more abundant yield meanwhile can avoid interferences from other resources. In addition, what is the main reason that the authors chosen ventral hair as control rather than performed the determination of chemical composition in neck mane samples during estrus and non-estrus seasons?
Response: Poll gland secretions are viscous liquids that exude through the skin covering the glands and adhere to the neck mane. Due to its large viscosity and weak fluidity, the collection effect with gauze is not significant, so the neck manes with Poll gland secretions are directly collected for determination. Thank you very much for your suggestions. Dominant males have their own herd, which is dominated by females. Other males are driven out of the herd. Therefore, in order to reduce the uncontrollable factors and individual differences in the experiment, the neck mane and ventral hair of the same camel were collected as controls, and the neck mane was not collected at different periods for detection. In addition, camels are seasonally estrus animals. If samples in estrus and non-estrus are to be collected, a long storage time will affect the determination of sample chemical composition.
Comment 3: In Line 24, it says “we determine the chemical composition of poll gland secretions” while in Line 28 it states, “to screen differentially expressed metabolites”. According to the text, the expression should be aligned to the latter one.
Response: Thank you for your suggestion. In this paper, the chemical composition of poll gland secretions was determined by comparing the characteristics of differentially expressed metabolites between the two groups. Therefore, differentially expressed metabolites could be considered as poll gland secretions.
Comment 4: This study aimed to screen metabolites and performed further analysis in relation to sexual behavior, however, little information on metabolites in poll gland was mentioned in the Introduction section. More literature related to the core content of this study is required.
Response: Thank you for your suggestion. We have revised it based on your suggestion.
Comment 5: Line 126-127, samples were collected from 6 male camels. If only one sample from each camel was collected and used for further GC/LC-MS metabolomic analysis, it could be too few samples for obtaining biologically acceptable results. The authors have to explain it carefully.
Response: Thank you for your suggestion. The samples used in this paper were from 6 male camels. Several neck mane samples and ventral hair samples were collected from each camel (each sample weight was more than 200 mg), and one sample was randomly selected for metabolomics detection, which was consistent with the minimum number of non-targeted metabolomics samples in animal tissues (biological replicates were no less than 6). In addition, internal standard and QC samples were used for quality control, and supervised orthogonal partial least squares-discriminant analysis (OPLS-DA) was used to improve the reliability of the experimental results.
Comment 6: Line 138, 191, 195, 206, the reference should be correctly inserted.
Response: Thank you for your suggestion. We have revised it based on your suggestion.
Comment 7: Line 405-406, a detailed discussion on the reason why in this study the typical sex pheromones were not detected is recommended.
Response: In the second part of the discussion section, the reasons why typical pheromones were not detected in this study are discussed in detail.
Comment 8: Combining the results of metabolites and related pathways, a definite conclusion on the molecular regulatory mechanism of poll gland secretion on camel estrus can be expected. Besides, it will be interesting to know the regulatory mechanism of poll gland appearance in estrus season and degradation thereafter.
Response: Your suggestions and prospects for the project are very good. We are also conducting relevant research now. Thank you for your recognition and interest in the project.

Round 2
Reviewer 1 Report
It is important that the authors follow the indications for writing the abstract of the article, which should be easy to understand for the readers, as well as adapt to the guidelines of the journal.
Author Response
Comment 1: It is important that the authors follow the indications for writing the abstract of the article, which should be easy to understand for the readers, as well as adapt to the guidelines of the journal.
Response: Thanks for your suggestion, we have reorganized the abstract following the writing instructions for the abstract. On behalf of my co-authors, we thank you for giving us a chance to revise and improve the quality of our article.
